# SCALABLE DIFFUSION FOR BIO-TOPOLOGICAL REPRESENTATION LEARNING ON BRAIN GRAPHS

## ABSTRACT

The topological structure information of the brain graph is critical in discovering bio-topological properties that underlie brain function and pathology. Authentic representations of brain graphs in many clinical applications heavily rely on these bio-topological properties. While existing studies have made strides in analyzing brain graph topology, they are often constrained by single-scale structural analysis and hence fail to extract these properties across multiple scales, thus potentially leading to incomplete and distorted representations. To address this limitation, we propose a novel Scalable diffusion model for bio-TOpological REpresentation learning on Brain graphs (BrainSTORE[1]). BrainSTORE constructs multiscale topological structures within brain graphs, facilitating a deep exploration of bio-topological properties. By embedding these features into the training process and prioritizing bio-topological feature reconstruction, BrainSTORE learns representations that are more reflective of underlying brain organization. Furthermore, BrainSTORE utilizes a unified architecture to integrate these features effectively, yielding improved bio-topological representations which are more robust and biologically meaningful. To the best of our knowledge, this is the first study to investigate bio-topological properties in brain graph representation learning. Extensive experiments demonstrate that BrainSTORE outperforms state-of-the-art methods in brain disease detection.

## 1 INTRODUCTION

Advanced multimodal neuroimaging data, such as diffusion tensor imaging (DTI) (Assaf & Pasternak, 2008) and functional magnetic resonance imaging (fMRI) (Van Den Heuvel & Pol, 2010), are used to construct structural and functional brain graphs, respectively (Peng et al., 2024). The topology of these multimodal brain graphs provides insights into the brain's *bio-topological properties*, including small-world, rich-club, and modular characteristics (Bullmore & Sporns, 2012). Learning representations from these properties deepens our understanding of the brain's complex organization, thereby supporting clinical diagnosis, cognitive impairment analysis, and the identification of new biomarkers (Tang et al., 2023; Yan et al., 2024).

Existing methods focus on learning brain graph representations from local connectivity or high-order structures (Safai et al., 2022; Zhu et al., 2022; Yang et al., 2023; Ye et al., 2024). While these approaches effectively analyze brain graph topology at a specific scale, recent studies emphasize the importance of examining bio-topological properties within modular and regional structures across multiple scales, particularly in neurological research (Fornito et al., 2015; Yan et al., 2024). Changes in these bio-topological properties are closely linked to neurological disorders such as Alzheimer's and Parkinson's, which often display disrupted small-world topology and decoupling of functional modules. These disruptions typically manifest as reduced global connectivity, reflecting significant alterations in brain organization (Liu et al., 2017; Zhang et al., 2024). Therefore, incorporating bio-topological properties across multiple scales in brain graph representation learning is essential for accurately presenting disease-related brain organization. However, the expressiveness of current methods is often constrained by single-scale structural analysis, limiting their ability to model multiscale brain graph topologies and leading to distorted representations.

---

[1] The codes are available at: `https://anonymous.4open.science/r/BrainSTORE-3CE9/`.

To address the above limitation, we propose a novel scalable diffusion model for *bio-topological representation learning* on brain graphs (**BrainSTORE**). Specifically, we introduce a *scalable learning strategy* designed to integrate remarkable bio-topological properties into the model training process, thereby enhancing the model's representation learning capabilities. This strategy features a novel multiscale community detection method inspired by brain hierarchy (Betzel & Bassett, 2017), which accounts for the structural dependencies of community assignments across various scales and modalities. This results in coherent and realistic multiscale topological structure partitions enabling detailed exploration of bio-topological properties. Additionally, recent studies have demonstrated the potential of diffusion models in representation learning (Yang et al., 2024a; Chen et al., 2024). We extend this approach to multimodal brain graph data by designing a unified architecture with modality-specific and shared backbone networks. BrainSTORE uniquely integrates bio-topological properties for *scalable joint denoising* and implementing a *scalable noise schedule* during diffusion. This enables BrainSTORE to prioritize the reconstruction of shared and complementary bio-topological features within multimodal brain graphs, facilitating the comprehensive capture and integration of these features. These advancements reduce potential biases associated with single-scale analysis and provide improved bio-topological representations that accurately reflect authentic brain organizational characteristics.

To summarize, our main contributions are three-fold: **1)** We propose a novel BrainSTORE model that pioneers the exploration of bio-topological properties in brain graph representation learning, effectively addressing representation distortion by delivering authentic bio-topological representations. **2)** We introduce a novel scalable learning strategy that models biologically realistic multiscale topological structures in brain graphs, enhancing the model expressiveness through integrating bio-topological properties into the training process. **3)** We conduct extensive experiments on multimodal brain disease datasets to validate the effectiveness of BrainSTORE, with additional explanation and ablation studies providing insights into the scalable diffusion mechanism.

## 2 RELATED WORK

### 2.1 BRAIN GRAPH REPRESENTATION LEARNING

Graph neural networks (GNNs) offer an effective approach for learning topological information from graph-structured data and have become widely utilized in modeling and representing brain network data (Bessadok et al., 2022). Cui et al. (2022) propose a unified brain graph representation learning framework. Similarly, BrainGNN (Li et al., 2021) further provides explainable biomarkers. Recently, most methods aim to integrate multimodal brain graph data to obtain improved representation. Simple methods extract topological features by applying GNNs to node connectivity and directly incorporate them (Zhu et al., 2022; Cai et al., 2022). Recently, approaches based on indirect interactions, such as Cross-GNN (Yang et al., 2023) and RH-BrainFS (Ye et al., 2024), have improved representations by considering the structural relationships across modalities, in which RH-BrainFS specifically extracting subgraph-level topological features. However, these methods often overlook bio-topological properties within multimodal graphs, limiting their expressiveness. In contrast, BrainSTORE detects multiscale topological structures across modalities to embed these properties into model training, achieving bio-topological representation learning.

### 2.2 DIFFUSION MODELS

Diffusion models are probabilistic generative models (Ho et al., 2020), which excel at learning flexible representations and are widely used in computer vision tasks (Preechakul et al., 2022; Yang & Wang, 2023). In particular, leveraging multimodal information from multiple tasks and data sources has proven effective for learning generalized representations. Current approaches can be divided into conditional models and multimodal models. Conditional models use modality embeddings to guide modality transformation, enabling tasks like text-to-video and text-to-image generation (Ma et al., 2023; Ho et al., 2022). Multimodal models, on the other hand, capture and generate data by integrating shared information across modalities such as MM-Diffusion (Ruan et al., 2023) and MT-Diffusion (Chen et al., 2024). Although these methods have succeeded, their use in graph learning is still restricted due to the inherent structural differences between images and graphs. Recently, DDM (Yang et al., 2024a) incorporates directional noise to capture meaningful semantic

and topological representations. However, DDM is tailored for mono-modal tasks, in contrast to BrainSTORE, which explores the application of diffusion models to multimodal graph data.

BrainSTORE is also related to community detection works discussed in Appendix A.

## 3 PRELIMINARIES

**Denoising Diffusion Probabilistic Models (DDPM)** is a Vallina diffusion model consisting of forward and reverse processes (Ho et al., 2020). In the forward process, Gaussian noise is incrementally added to the original data point $\mathbf{x} \sim q(\mathbf{x})$ following a Markov chain until it transforms into isotropic white noise $\mathcal{N}(0, \mathbf{I})$. The reverse process uses a neural network to remove the noise and restore the data to its original distribution. Mathematically, the forward process from step $t-1$ to $t$ is defined as: $q(\mathbf{z}_t|\mathbf{z}_{t-1}) = \sqrt{1-\beta_t}\mathbf{z}_{t-1} + \sqrt{\beta_t}\epsilon$, where $\epsilon \in \mathcal{N}(0, \mathbf{I})$, $\mathbf{z}_t$ is the noisy representation, $\beta_t$ controls the noise level, and $\epsilon$ is Gaussian noise. The reverse process then iteratively denoises $\mathbf{z}_T$ back to its initial state, recovering the original representation $\mathbf{z}_0$. With both processes generating a sequence of noisy representations $\mathbf{z}_0, \ldots, \mathbf{z}_T$, the model is optimized by minimizing the variational lower bound loss: $\mathcal{L} := \mathbb{E}_{t,\mathbf{z}_0,\epsilon}\left[\|\epsilon - \epsilon_\theta(\sqrt{\overline{\alpha}_t}\mathbf{z}_0 + \sqrt{1-\overline{\alpha}_t}\epsilon, t)\|^2\right]$, where $\overline{\alpha}_t = \prod_1^t(1-\beta_t)$, and $\epsilon_\theta(\cdot)$ represents the denoising model, typically structured as a U-Net (Ronneberger et al., 2015).

**Problem Definition.** Given a brain network dataset for $M$ subjects, $\{\mathcal{G}_1, \mathcal{G}_2, \ldots, \mathcal{G}_M\}$, where each $\mathcal{G} = (G^{\mathrm{sc}}, G^{\mathrm{fc}})$ represents multimodal brain graph data comprising both structural and functional graphs constructed from DTI and fMRI data, respectively. Each modality of the brain graph is represented as $G = (V, \mathbf{A}, \mathbf{X})$, where $V$ is a finite set of nodes with size $N$, $\mathbf{A} \in \mathbb{R}^{N \times N}$ is the adjacency matrix, and $\mathbf{X} \in \mathbb{R}^{N \times N}$ is the node feature matrix. The nodes represent the regions of interest (ROIs) in brain networks, and the connectivity strengths between paired ROIs are defined as the elements $a_{ij} \in \mathbf{A}, (i, j = 0, \ldots, N)$. The connectivity correlation vector is the node feature vector, $\mathbf{x}_i \in \mathbb{R}^N$. The objective is to learn a network $f_\theta(\cdot, \cdot)$, with a series of scale resolutions $\{\lambda_{\min}, \ldots, \lambda_{\max}\}$, that capable of encoding multimodal brain graph features into bio-topological representations $\mathbf{Z} = [\mathbf{z}_1, \ldots, \mathbf{z}_N] \in \mathbb{R}^{N \times D}$, where $\mathbf{z}_n \in \mathbb{R}^D$ represents the feature vector for the $n$-th node. These representations are then utilized for graph classification tasks.

## 4 DESIGN OF BRAINSTORE

This section introduces our novel BrainSTORE, depicted in Figure 1, designed to enhance diffusion models for multimodal graph data by incorporating structural topological attributes across modalities. This approach facilitates the bio-topological representation learning on brain graph. We begin by detailing the strategy designed to improve the model's representation learning capabilities in Section 4.1. Next, Section 4.2 discusses model training, which leverages this strategy and presents the scalable joint denoising process within our unified model. Section 4.3 outlines a scalable noise schedule tailored for this denoising process. Finally, we address the bio-topological representation learning from the denoising model in Section 4.4.

### 4.1 SCALABLE LEARNING STRATEGY

Inspired by the hierarchical nature of brain graphs (Betzel & Bassett, 2017), we propose a novel scalable learning strategy to model multiscale topological structures within brain graphs through a community detection method. However, this poses two primary challenges when implemented in multimodal brain graphs: First, the hierarchical structure often results in brain regions displaying stable allegiance across scales, where the communities at adjacent scales influence node assignments at a given scale (Uddin et al., 2019; Vaiana & Muldoon, 2020). Second, due to the solid structural coupling between structural and functional brain graphs (Amico & Goñi, 2018), where community assignments at the same scales exhibit high correlations, especially at intermediate topological scales (Ashourvan et al., 2019). Nevertheless, traditional community detection methods independently identify communities for each modality or scale, leading to potential inconsistencies.

To address these challenges, our method extends the traditional Louvain algorithm (Blondel et al., 2008) by optimizing the dependencies within and between scales across different modalities. Specifically, it integrates a multiscale connection parameter $\tau$ and a multimodal connection parameter $\kappa$

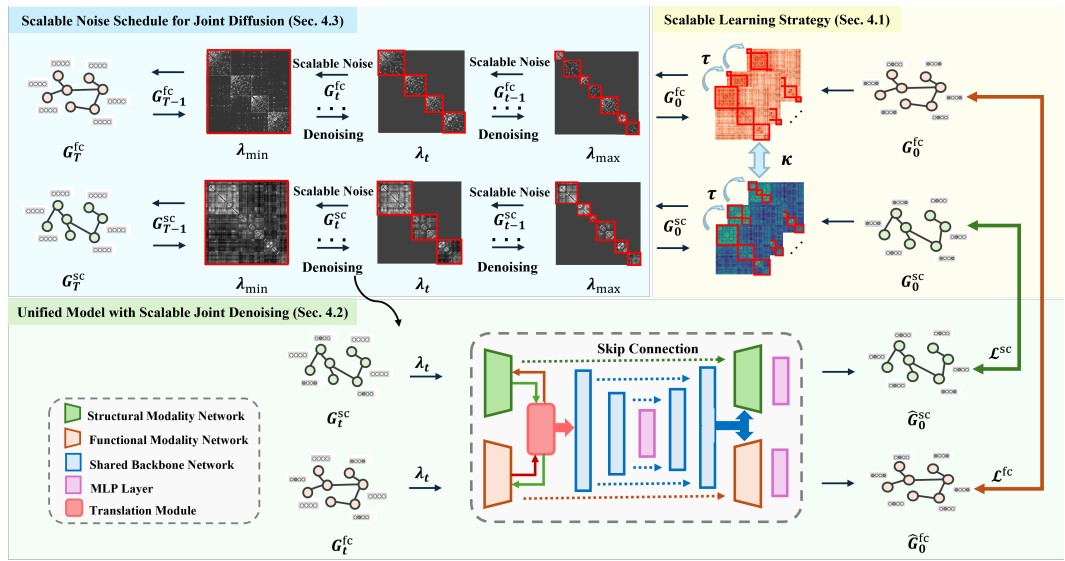

Figure 1: Overview of the BrainSTORE framework: The *Scalable Learning Strategy* (yellow) detects multiscale topological structures in multimodal brain graphs. These findings guide the *Scalable Noise Schedule* (blue) for performing joint diffusion on brain graph data. The unified model architecture performs *Scalable Joint Denoising* (green), enabling the adaptive reconstruction of biotopological features in original brain graph data.

to detect topological structures across multiple scales in brain graphs. This approach effectively ensures a more coherent and integrated representation of the brain organization, revealing biotopological properties.

As shown in Figure 2, $\tau$ adjusts the dependencies of community assignment across adjacent scales within each modality. This facilitates the gradual decomposition of large communities in the initial layer $l_1$ (with minimal scale resolution parameter $\lambda_{l_1}$) into smaller communities in adjacent layers $l_2$ (with a linearly increased scale resolution parameter $\lambda_{l_2}$). For parameter $\kappa$, it establishes dependencies between nodes across different modalities at the same scale. Formally, with the community detection of layer $l_1$ in each modality graph, we define the scalable quality function for layer $l_2$ in $m_1$ modality graph as follows:

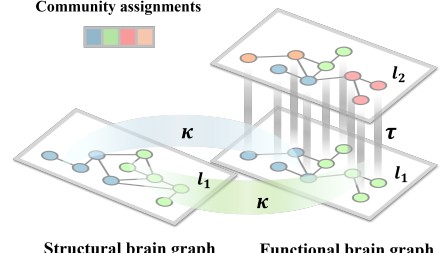

Figure 2: Schematic representation of multiscale community detection

$$
Q_{\tau,\kappa}(\lambda) = \frac{1}{2\eta} \sum_{ij|l_1l_2m_1m_2} \Big\{ \delta_{(l_1m_1,l_2m_1)} \big( a_{ij|l_1m_1} - \lambda_{l_1} p_{ij|l_1m_1} \big)
$$
$$
+ \delta_{(i|l_1,j|l_2)} \tau_{j|l_1l_2m_1} + \delta_{(i|m_1,j|m_2)} \kappa_{j|l_1m_1m_2} \Big\} \delta(c_{i|l_1m_2}, c_{j|l_2m_1}),
\tag{1}
$$

where $a_{ij|l_1m_1}$ and $p_{ij|l_1m_1}$ represent the connection strength and expected strength between nodes $i$ and $j$ in layer $l_1$, $\tau_{j|l_1l_2m_1}$ gives the connection strength from node $j$ in layer $l_1$ to layer $l_2$ within $m_1$ modality graph, and the $\kappa_{j|l_1m_1m_2}$ indicates the strength of the connections for node $j$ at layer $l_1$ across modalities. The total edge weight in the $m_1$ modality graph denotes as $\eta = \frac{1}{2} \sum_{j|l_2m_1} (A_{j|l_2m_1} + T_{j|l_2m_1} + K_{j|l_2m_1})$, where $A_{j|l_2m_1} = \sum_i a_{ij|l_2m_1}$ and $T_{j|l_2m_1} = \sum_{l_1} \tau_{j|l_1l_2m_1}$ are the sum of intra-layer and inter-layer connection strength of $j$-th node, and $K_{j|l_2m_1} = \sum_{m_1} \kappa_{j|l_1m_1m_2}$ is the sum of connection strength for layer $l_1$ across modalities. Here, the Kronecker delta function $\delta(c_{i|l_1m_2}, c_{j|l_2m_1})$ returns 1 if node $i$ (layer $l_1$, modality $m_2$) and the node $j$ (layer $l_2$, modality $m_1$) belong to the same community ($c_{i|l_1m_2} = c_{j|l_2m_1}$), and 0 otherwise. As $\tau$ increases, nodes across adjacent scales exhibit stronger structural dependencies, creating

a hierarchical organization where smaller topological structures are nested within larger ones. Similarly, adjusting $\kappa$ modifies the similarity of these structures between modalities: a lower $\kappa$ value leads to more independent structural partitions, whereas a higher $\kappa$ enhances structural consistency. Detailed parameter settings are provided in Appendix B.1. Finally, the bio-topological properties in these identified structures will be integrated into the joint denoising and diffusion processes for bio-topological representation learning on brain graphs.

## 4.2 UNIFIED MODEL WITH SCALABLE JOINT DENOISING

We develop a unified denoising model, $f_\theta(\cdot, \cdot)$, tailored to extract bio-topological features from multimodal brain graphs by integrating the key bio-topological properties into model training. Our model diverges from traditional approaches that denoise the entire graph. Instead, it focuses on joint denoising within identified multiscale topological structures, progressing from coarse ($\lambda_{\min}$) to fine ($\lambda_{\max}$) resolutions. This scalable approach effectively allows the model to prioritize reconstructing bio-topological features of original brain graph data.

**Learning Objectives.** We follow the standard training protocol for diffusion models, performing denoising during the reverse process. Our method uniquely employs scalable joint denoising, adapting the scale resolution parameter ($\lambda_{\min}, \ldots, \lambda_{\max}$) as defined in Equation 1 dynamically for using the defined topological structure as a prior during denoising. Specifically, for each modality, we define the reverse process as $p_{f_\theta}(G_{t-1}|G_t) = \mathcal{N}(G_{t-1}; \mu_Q(G_t^{\mathrm{sc}}, G_t^{\mathrm{fc}}; Q(\lambda_t), t))$, where $Q(\lambda_t)$ delineates the topological structures at the $t$-th step. Since the learning objective is targeted at multimodal data, this setup requests for the generation of $G_{t-1}$ from a Gaussian distribution jointly informed by the correlation between $G_t^{\mathrm{sc}}$ and $G_t^{\mathrm{fc}}$. However, directly optimizing $p_{f_\theta}(\cdot)$ using the variational lower bound is often unstable and requires various optimization techniques for stabilization. Drawing from (Li et al., 2022; Bansal et al., 2024), we adopt an alternative objective where the denoising model $f_\theta(G_t^{\mathrm{sc}}, G_t^{\mathrm{fc}}; Q(\lambda_t), t)$ directly predicts $G_0^{\mathrm{sc}}$ and $G_0^{\mathrm{fc}}$. Thus, the optimization objective function is formulated as follows:

$$\mathcal{L} = \mathbb{E}_{\mathbf{X}_0^{\mathrm{sc}}, \mathbf{X}_0^{\mathrm{fc}}, t} \left[ \|f_{\theta_{\mathrm{sc}}}(\mathbf{A}^{\mathrm{sc}}, \mathbf{X}_t^{\mathrm{sc}}; Q(\lambda_t), t) - \mathbf{X}_0^{\mathrm{sc}}\|^2 + \|f_{\theta_{\mathrm{fc}}}(\mathbf{A}^{\mathrm{fc}}, \mathbf{X}_t^{\mathrm{fc}}; Q(\lambda_t), t) - \mathbf{X}_0^{\mathrm{fc}}\|^2 \right], \quad (2)$$

where $\mathbf{X}_t^{\mathrm{sc}}$ and $\mathbf{X}_t^{\mathrm{fc}}$ are the noisy brain graph data at step $t$ for each modalities. Meanwhile, $\mathbf{X}_0^{\mathrm{sc}}$ and $\mathbf{X}_0^{\mathrm{fc}}$ represent the corresponding original data. The function $f_{\theta_{\mathrm{sc}}}(\cdot)$ and $f_{\theta_{\mathrm{fc}}}(\cdot)$ are modality-special networks of our unified denoising model, tailored to handle structural and functional data respectively. By encoding a series of bio-topological features, our model strives to minimize noise and enhance the fidelity of the brain graph data reconstructed at each reverse step.

**Model Architecture.** To parameterize the denoising model, we introduce a symmetric architecture inspired by U-Net, which features a shared backbone network alongside modality-specific networks (functional and structural), as depicted in Figure 1. The shared backbone network is the central hub for integrating and processing information from both modality-specific networks, providing shared bio-topological features. Each modality-specific network operates independently, focusing on encoding modality complementary bio-topological features. This dual-branch structure allows our model to perform joint denoising within defined structures, comprehensively enhancing the integration of bio-topological features across multimodal brain graphs.

Each network consists of several GNN layers and multilayer perceptrons (MLPs), organized into down-sampling, mid-sampling, and up-sampling blocks. Initially, modality-specific networks use GNN layers as down-sampling blocks to encode the input noisy graphs within the topological structure with resolution $\lambda_t$ into low-dimensional embeddings, formulated as $\mathrm{GNN}(\mathbf{A}_t, \mathbf{X}_t; Q(\lambda_t), t)$. The obtained embeddings $\mathbf{H}_t^{\mathrm{sc}}, \mathbf{H}_t^{\mathrm{fc}} \in \mathbb{R}^{N \times d_h}$ are then utilized for joint denoising through the shared backbone network, which processes a coupled graph $G_t^{\mathrm{cp}} = (V, \mathbf{A}_t^{\mathrm{cp}}, \mathbf{X}_t^{\mathrm{cp}})$, constructed using a linear GNN-based translation module $\mathrm{Tran}(\cdot, \cdot)$. This involves translating from functional to structural brain graphs as $\mathbf{H}_t^{\mathrm{sc}\prime} = \mathrm{Tran}_{\mathrm{fc} \to \mathrm{sc}}(\mathbf{H}_t^{\mathrm{sc}}, \mathbf{H}_t^{\mathrm{fc}})$, followed by a bidirectional translation for $\mathbf{H}_t^{\mathrm{fc}\prime} = \mathrm{Tran}_{\mathrm{sc} \to \mathrm{fc}}(\mathbf{H}_t^{\mathrm{sc}\prime}, \mathbf{H}_t^{\mathrm{fc}})$, and vice versa. By optimizing the bidirectional translation loss, $\|\mathbf{H}_t^{\mathrm{fc}} - \mathbf{H}_t^{\mathrm{fc}\prime}\|^2 + \|\mathbf{H}_t^{\mathrm{sc}} - \mathbf{H}_t^{\mathrm{sc}\prime}\|^2$, we define the optimized linear GNN matrix as the $\mathbf{A}^{\mathrm{cp}}$ representing their structural correlation. The feature matrix $\mathbf{X}^{\mathrm{cp}}$ is defined as $\frac{1}{2} \left[ \mathbf{H}_t^{\mathrm{fc}}(\mathbf{H}_t^{\mathrm{sc}})^\top + \mathbf{H}_t^{\mathrm{sc}}(\mathbf{H}_t^{\mathrm{fc}})^\top \right] \in \mathbb{R}^{N \times N}$, integrating modality-specific features. The shared backbone network processes this coupled graph through additional GNN layers and an MLP to encode shared bio-topological features, outputting the coupled embedding $\mathbf{H}_t^{\mathrm{cp}}$. Skip connections are intro-

duced here to prevent over-smoothing and enhance information retention. Ultimately, the modality-specific networks leverage embeddings processed from both the initial downsampling blocks and the shared backbone network for reconstructing complementary bio-topological features, formulated as $\text{GNN}(\mathbf{A}_t, \mathbf{H}_t + \mathbf{H}_t^{\text{cp}}; Q(\lambda_t), t)$. This process ensures that the reconstructed brain graph data, denoted as $\hat{\mathbf{X}}_0^{\text{sc}}$ and $\hat{\mathbf{X}}_0^{\text{fc}}$, each in $\mathbb{R}^{N \times N}$, reflect a refined synthesis of modality-specific and cross-modality insights enhancing the accuracy and robustness of the denoising output $\hat{G}_0^{\text{sc}}$ and $\hat{G}_0^{\text{fc}}$.

### 4.3 Scalable Noise Schedule for Joint Diffusion

In a standard diffusion framework, the reverse process reconstructs the Gaussian noise added during the forward process. Building on this, we aim to generate scalable noise under key bio-topological properties for performing joint diffusion on multimodal brain graphs corresponding to the denoising process. Given graph data's unique anisotropic and directional structures, we introduce directionality as a constraint in the noise generation process inspired by the DDM. Specifically, we define each modality of the noisy brain graph at $t$-th forward step as $G_t = (V, \mathbf{A}, \mathbf{X}_t)$, where $\mathbf{X}_t$ represent the noisy feature representations. Taking the forward process of $i$-th node features $\mathbf{x}_{t,i} \in \mathbb{R}^N$ from step $t-1$ to step $t$ in each modality as an example, it can be formulated as:

$$q(\mathbf{x}_{t,i}|\mathbf{x}_{(t-1),i}) = \sqrt{1-\beta_t}\mathbf{x}_{(t-1),i} + \sqrt{\beta_t}\hat{\epsilon}, \tag{3}$$

$$\hat{\epsilon} = \text{sgn}(\mathbf{x}_{0,i}; Q(\lambda_{t-1})) \odot |\mu_Q + \sigma_Q \odot \epsilon|, \tag{4}$$

where $\mathbf{x}_{0,i}$ is the original node feature vector, and $\mu_Q$ and $\sigma_Q$ denote the mean and standard deviation values of the node features in the identified structure at scale resolution $\lambda_{t-1}$. The symbol $\odot$ represents the Hadamard product. Equation 4 transforms data-agnostic Gaussian noise into anisotropic noise, incorporating its correlation within the data batch. Notably, in the direction function $\text{sgn}(\cdot)$, we introduce the topological structures identified by the quality function in Equation 1 as a conditioning factor. Unlike the DDM, which calculates node direction across the entire batch of graphs, we focus on nodes within the same topological structure in this batch to compute their direction, along with the shared empirical mean and standard deviation. This approach restricts the forward diffusion process to the topological structures within the batch, preventing excessive divergence and ensuring consistency in topological features. Consequently, the reverse process yields a series of noisy multimodal graphs at various scales, ranging from fine ($\lambda_{\text{max}}$) to coarse ($\lambda_{\text{min}}$) granularity. The detailed training algorithm is summarized in Algorithm 1 in Appendix C.

### 4.4 Representation Learning

For a single subject with multimodal graph data $G^{\text{sc}}$ and $G^{\text{fc}}$, the bio-topological representations are derived from the activations at selected $K$ time steps within the denoising model. Specifically, we leverage the activations from the final upsampling blocks of each modality because these layers encode both modality-specific and shared bio-topological features effectively. The scale resolutions vary across these $K$ denoising steps, which is crucial to ensure that the representations reflect the bio-topological properties accurately across different scales. The embeddings from these activations are represented as $\hat{\mathbf{H}} \in \mathbb{R}^{N \times d_h}$ for each modality. At each selected $k$-th step, these embeddings are concatenated to form the multimodal representations, expressed as $\mathbf{z}_k = [\hat{\mathbf{H}}_k^{\text{sc}}, \hat{\mathbf{H}}_k^{\text{fc}}] \in \mathbb{R}^{N \times 2d_h}$. The comprehensive bio-topological representations are then assembled by aggregating all $K$-step representations into $\mathbf{Z} = [\mathbf{z}_1, \mathbf{z}_2, \ldots, \mathbf{z}_K] \in \mathbb{R}^{N \times D}$, where $D = K \times 2d_h$. This methodology ensures that the learned representations integrate detailed bio-topological features, enhancing the model's efficacy in disease detection through graph classification. The detailed steps of this representation learning process are outlined in Algorithm 2 in Appendix C.

## 5 Experiments

### 5.1 Experimental Settings

**Datasets.** We evaluate our BrainSTORE method on two real-world medical datasets. 1) The Alzheimer's Disease Neuroimaging Initiative (ADNI) dataset, used for diagnosing Alzheimer's disease (AD) progression, which includes 54 AD samples, 195 mild cognitive impairment (MCI) samples, and 211 normal control (NC) samples, categorized according to standard clinical criteria. 2)

Table 1: Accuracy (%) on the ADNI and PPMI datasets ("FC" and "SC" are the functional and structural modality, respectively).

| Methods | Modality | ANDI | | | PPMI |
|---|---|---|---|---|---|
| | | NC vs MCI | MCI vs AD | NC vs AD | HC vs PD |
| BrainGNN | SC | $54.3 \pm 9.4$ | $57.2 \pm 14.7$ | $61.7 \pm 8.5$ | $61.5 \pm 12.5$ |
| BrainGNN | FC | $51.8 \pm 4.3$ | $71.1 \pm 3.7$ | $60.0 \pm 7.7$ | $64.6 \pm 18.4$ |
| BrainGB | SC | $55.8 \pm 4.8$ | $74.5 \pm 6.9$ | $73.9 \pm 7.3$ | $66.8 \pm 7.3$ |
| BrainGB | FC | $56.6 \pm 2.5$ | $77.3 \pm 4.7$ | $61.4 \pm 6.5$ | $68.4 \pm 10.7$ |
| TAN | SC,FC | $71.5 \pm 10.3$ | $81.2 \pm 7.2$ | $75.2 \pm 9.8$ | $75.1 \pm 8.5$ |
| Cross-GNN | SC,FC | $\underline{82.8 \pm 6.3}$ | $83.4 \pm 6.1$ | $80.3 \pm 8.1$ | $84.6 \pm 7.1$ |
| RH-BrainFS | SC,FC | $80.4 \pm 7.4$ | $\underline{85.3 \pm 5.9}$ | $\underline{82.4 \pm 7.9}$ | $\underline{85.6 \pm 7.1}$ |
| **BrainSTORE** (ours) | SC,FC | $\mathbf{85.3 \pm 6.4}$ | $\mathbf{89.4 \pm 4.7}$ | $\mathbf{90.9 \pm 4.9}$ | $\mathbf{88.9 \pm 3.4}$ |

The Parkinson's Progression Markers Initiative (PPMI) dataset, used for diagnosing Parkinson's disease (PD), contains 41 healthy controls (HC) and 49 PD patients. Detailed information on the datasets and their preprocessing can be found in Appendix D.1 and Appendix D.2.

**Metrics.** To ensure fairness, we evaluate all methods using 10-fold cross-validation with the same training and testing dataset splits. We use the mean and standard deviation of the following metrics to assess the classification performance: accuracy (ACC), sensitivity (SEN), specificity (SPE), F1-score, and the area under the ROC curve (AUC).

**Implementation Details.** For all experiments, we use the Adam optimizer with an initial learning rate of $e^{-4}$ and a dropout rate of 0.2, training for 100 epochs. In the BrainSTORE model, we set the multiscale and multimodal connection parameters, $\tau$ and $\kappa$, to 0.5 and 1.0. The multiscale resolutions $\lambda$ are set to $[0.5, 1.5]$, the number of denoising step $K$ is set to 3, and the shared backbone network in the U-Net architecture includes 4 GNN layers, with each layer having 4 attention heads. Our experiments are implemented in PyTorch and trained on an NVIDIA 3090 GPU.

## 5.2 COMPARISON EXPERIMENTS

**Baselines.** We select state-of-the-art brain graph representation learning methods as baselines, categorized into mono-modal and multimodal approaches. For mono-modal methods, we evaluate BrainGNN (Li et al., 2021) and BrainGB (Cui et al., 2022) using structural and functional brain graphs. For multimodal methods, we include TAN (Zhu et al., 2022), Cross-GNN (Yang et al., 2023), and RH-BrainFS (Ye et al., 2024), and test these methods on the same datasets used for our model evaluation. All baseline implementations are conducted using the original code from their respective publications.

**Results and Analysis.** Table 1 shows that our model outperforms others in the ACC metric across all datasets. Multimodal methods generally exceed mono-modal ones by leveraging topological information from both modalities. In the ADNI dataset, we achieve a 5.0% average improvement over other multimodal baselines, with a 3.3% improvement in the PPMI dataset. Notably, our model demonstrates an 8.5% increase in the NC and AD comparison group, attributed to our scalable learning strategy's effective identification of significant topological structure differences (as confirmed in Section 5.4). These bio-topological features robustly represent authentic brain graph data. Additionally, BrainSTORE shows improvements in other metrics, with detailed results in Appendix E.

**Visualization.** We visualize the results of multimodal methods on the ADNI dataset to showcase their capability in brain graph representation learning. We use t-SNE to visualize graph-level embeddings from each method's final layer. As Figure 3 illustrates, TAN and Cross-GNN formed clusters that did not separate the classes. While RH-BrainFS showed a similar pattern to our method, it still displayed significant overlap at class boundaries. In contrast, our method effectively minimized overlap, resulting in clearer class distinctions.

## 5.3 ABLATION STUDY

**Impact of Scalable Learning Strategy.** We assess the effectiveness of the scalable learning strategy through several metrics: 1) Multiscale community detection methods: We replace it with several re-

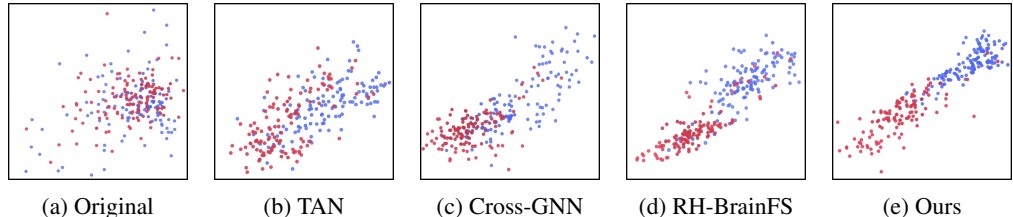

(a) Original     (b) TAN     (c) Cross-GNN     (d) RH-BrainFS     (e) Ours

Figure 3: Visualization of classification results for multimodal methods on the ADNI dataset.

cent algorithms, including the Girvan-Newman (Despalatović et al., 2014), spectral clustering (Newman, 2013), and the Louvain algorithm (Zhang et al., 2021). As shown in Figure 4, our method outperforms these existing approaches. This demonstrates our method's potential and advantages in capturing the complex structure of brain networks, highlighting the importance of exploring community assignment dependencies across different modalities and scales.

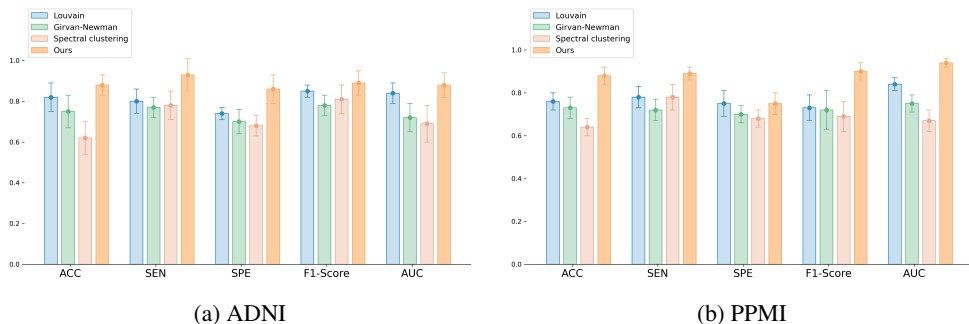

(a) ADNI                 (b) PPMI

Figure 4: Performance of the model using the scalable learning strategy designed with different community detection methods.

2) Model comparison under different diffusion time steps: We evaluate model performance trained with scalable and white noise schedules at each reverse step, as shown in Figure 5. The results indicate that when combined with the scalable learning strategy, the model enhances the quality of bio-topological representation learning. Although performance may decline at longer time steps due to sparser perturbation sampling and increased information-sharing complexity, it still effectively retains essential information for downstream tasks.

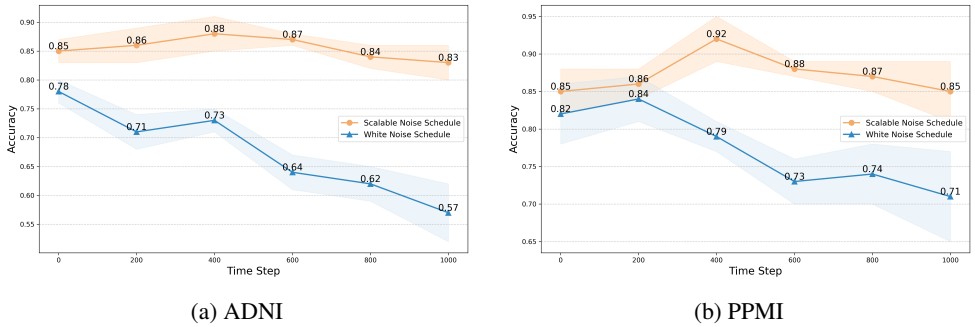

(a) ADNI                 (b) PPMI

Figure 5: Performance of the model trained using scalable and white noise schedules under different diffusion time steps.

**Effectiveness of Main Modules.** In this section, we evaluate the effectiveness of our model's architecture, as shown in Table 2. 1) Modality-specific network: We assess the impact of using only shared backbone network embeddings. The results indicate that jointly learning from both networks effectively provides shared and complementary information across multimodal brain graphs.

2) Skip connections: We analyze the effect of removing skip connections between upsampling and downsampling layers. Although this did not significantly decrease performance, it increased result variance, suggesting skip connections stabilize model performance. 3) Scalable learning strategy: We test this strategy by varying its activation during training. Results show the effects of removing the strategy in forward and reverse processes, indicated by "w/o F" and "w/o R," respectively, while "w/o F&R" denotes removal in both processes. This highlights the benefits of embedding bio-topological properties in brain graph representation learning, particularly in the forward process.

Table 2: Performance of main modules.

| Modules | ACC | SEN | SPE | F1-score | AUC |
|---|---|---|---|---|---|
| w/o Modality-specific network | $85.5 \pm 7.0$ | $82.4 \pm 5.8$ | $84.7 \pm 3.8$ | $88.9 \pm 4.7$ | $85.9 \pm 6.8$ |
| w/o Skip connections | $86.5 \pm 8.5$ | $87.9 \pm 8.2$ | $84.2 \pm 9.6$ | $87.9 \pm 10.9$ | $80.7 \pm 8.8$ |
| w/o F | $78.8 \pm 6.3$ | $82.9 \pm 5.7$ | $79.4 \pm 6.9$ | $80.5 \pm 8.7$ | $83.2 \pm 7.6$ |
| w/o R | $82.4 \pm 7.9$ | $86.6 \pm 7.2$ | $84.9 \pm 8.4$ | $87.5 \pm 5.9$ | $83.6 \pm 8.4$ |
| w/o F&R | $75.5 \pm 7.0$ | $79.6 \pm 6.8$ | $73.7 \pm 8.5$ | $75.9 \pm 7.4$ | $73.2 \pm 5.8$ |
| **BrainSTORE** (ours) | $\mathbf{88.6 \pm 4.8}$ | $\mathbf{92.1 \pm 6.7}$ | $\mathbf{83.1 \pm 6.3}$ | $\mathbf{89.6 \pm 5.6}$ | $\mathbf{89.9 \pm 7.4}$ |

## 5.4 Explanation of Scalable Learning Strategy

This strategy identifies multiscale topological structures in brain graphs, which can provide insights into neurological disorders. To validate this, we present visualizations of identified multiscale topological structures under a series of resolution parameters $\lambda$ on functional brain graphs in Figure 6. Specifically, the AD group has fewer large communities at the small-scale resolution than the NC group, while the number of communities increases at the larger scale. This indicates changes in the brain's overall topological structure, leading to reduced isolation between functional networks and larger co-classifications during structure partitions, which aligns with current medical research (Contreras et al., 2019). Notably, brain graphs in the MCI stage reveal that as clinical symptoms worsen, connections between multiscale communites become more intertwined, highlighting a clear continuity in AD progression. More visualization results can be found in Appendix B.2.

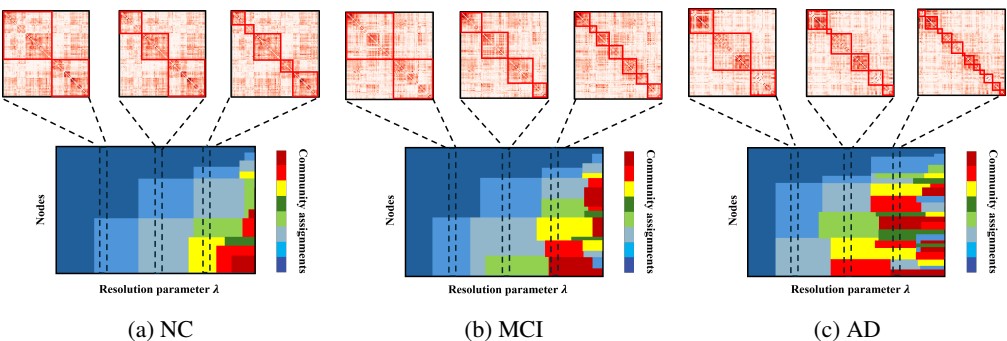

(a) NC          (b) MCI          (c) AD

Figure 6: Visualization of multiscale community detection results on the ADNI dataset.

## 6 Discussions

**Conclusion.** This paper introduces BrainSTORE, a novel brain graph representation learning model that addresses the limitations of existing methods in capturing bio-topological representations. By integrating a scalable learning strategy, BrainSTORE embeds bio-topological properties into model training, enhancing its representation learning capabilities. It effectively captures and integrates topological information from multimodal brain graphs within a unified framework, yielding improved bio-topological representations. Our results demonstrate that BrainSTORE outperforms state-of-the-art methods in disease detection tasks, confirming that the learned representations accurately reflect authentic brain characteristics.

**Limitations and Future Work.** Due to the scarcity and complexity of medical data acquisition and processing, the available datasets are limited, which may introduce bias in model learning. In future work, we will focus on collecting more high-quality data to mitigate this issue. Additionally, while our model introduces bio-topological representation learning, it does not explain the relationship between bio-topological features and corresponding structures. Moving forward, we will explore these explanations and investigate additional bio-topological features.

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

## A  RELATED WORK

### A.1  BRAIN GRAPH COMMUNITY DETECTION

Community detection is essential in brain graph analysis, focusing on identifying clusters within different brain regions to improve our understanding of the brain's organization and function. Most research has concentrated on single-scale communities, using algorithms like the Girvan-Newman method and modularity optimization (Garcia et al., 2018; Sporns & Betzel, 2016). Recent advancements have introduced more complex techniques, such as spectral clustering (Liang et al., 2019) and hierarchical clustering (Ashourvan et al., 2019), which capture the intricate relationships in brain networks. For example, spectral clustering uses eigenvalues of the adjacency matrix to identify topological structures, while hierarchical methods enable multiscale detection, revealing the layered organization of brain regions. Yang et al. (2024b) propose a transformer-based method novelty takes the community detection as a token clustering task. Despite these advances, our understanding is still limited due to the complexities of networks across scales and modalities (Betzel & Bassett, 2017). Thus, we propose a new multiscale community detection method that considers both hierarchical structure and structural relationships across different modalities.

## B  IMPLEMENTATION OF SCALABLE LEARNING STRATEGY

### B.1  PARAMETER SETTINGS

In this section, we discuss the settings for the multiscale connectivity parameter $\tau$ and the multimodal connectivity parameter $\kappa$ within the quality function. Figure 7 illustrates the community partition results of brain graph nodes at different $\tau$ settings. As $\tau$ increases, the correlations between hierarchical topological structures become more pronounced, indicating a stronger dependence of node allocation across scales. However, excessively high parameter values can complicate community detection, as shown in Figure 7c, where larger resolution parameters are required for effective partitioning. Ultimately, we determined the optimal parameter $\tau$ value to be $0.5$.

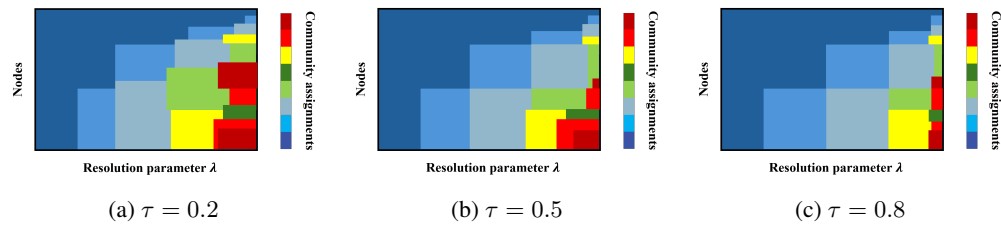

(a) $\tau = 0.2$       (b) $\tau = 0.5$       (c) $\tau = 0.8$

Figure 7: Visualization of community detection results under different multimodal connectivity parameter settings.

To evaluate the effectiveness of our multimodal connectivity parameters, we adjusted the parameter $\kappa$ to values of $0.2, 0.5, 0.8$, and $1.0$ under the mid-scale connectivity settings, where the community correlation across modalities is most obvious. We assessed the community detection results for the same nodes across different modality brain graphs using community label differences and community assignment similarity. As shown in Figure 9, it is evident that as $\kappa$ increases, the community assignments for the same nodes across different modalities become more consistent. To extract the most representative topological structure partitions, we ultimately set $\kappa$ to $1.0$.

Additionally, we discuss the range of resolution parameter settings for multiscale community detection, where the complete sample set varies from the minimum setting (with a community count of one) to the maximum setting (where the community count equals the number of nodes). Combining these parameters and leveraging existing knowledge of brain functional modules, we identified an optimal set of multiscale community resolutions, $\lambda = [0.5, 1.5]$, to ensure clear topological structure delineation and coherence.

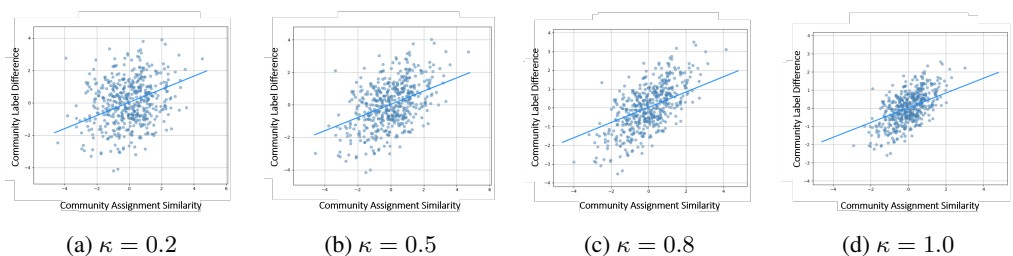

(a) $\kappa = 0.2$  (b) $\kappa = 0.5$  (c) $\kappa = 0.8$  (d) $\kappa = 1.0$

Figure 8: Community detection results under different multimodal connectivity parameter settings.

## B.2 VISUALIZATION

We present schematic representations of multiscale topological structures detected in the structural and functional brain graphs of various populations from the ADNI and PPMI datasets based on the established multiscale community parameters in Figure 8. The results highlight the consistency of community partitioning in multimodal brain graphs and the correlations across different scales. Additionally, communities in both PD and AD patients are more dispersed, with a higher quantity of smaller communities. This observation aligns with current medical research indicating that these conditions often exhibit reduced global connectivity and decoupling of functional modules, further reinforcing the validity of our findings (Fornito et al., 2015; Liu et al., 2017).

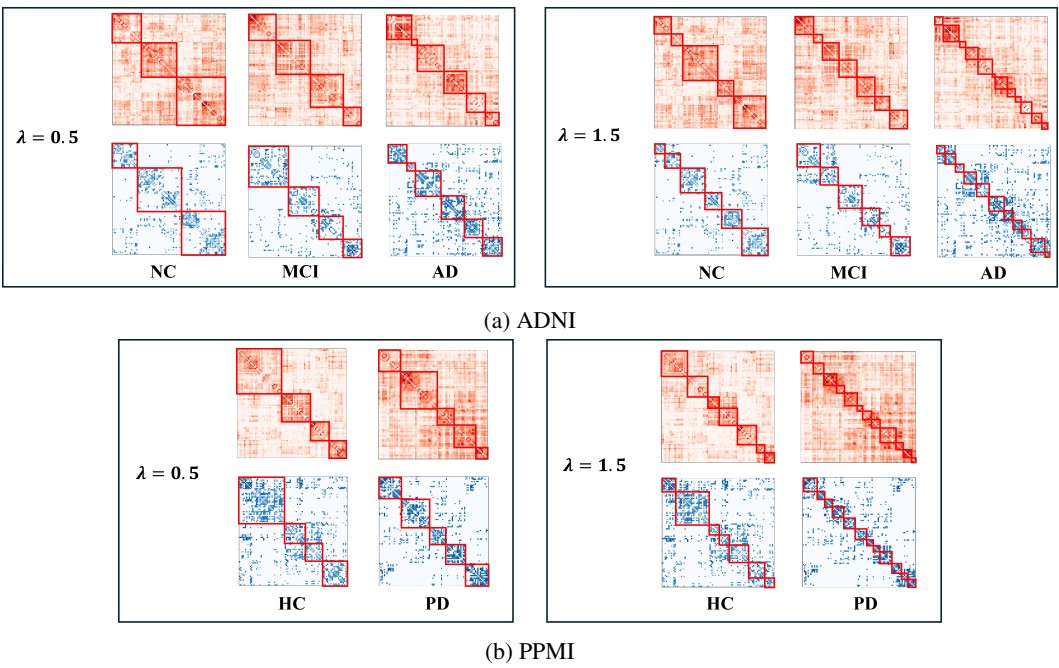

(a) ADNI

(b) PPMI

Figure 9: Visualization of multiscale community detection results on sturctual (blue) and functional (red) brain graphs.

## C THE COMPLETE ALGORITHM

This section presents the complete algorithm for our proposed scalable diffusion model.

---

**Algorithm 1** The training algorithm.

---

1: **Input:** A batch of brain graph datasets $\{\mathcal{G}_1, \ldots, \mathcal{G}_B\}$, $\mathcal{G} = (G^{\mathrm{sc}}, G^{\mathrm{fc}})$, a series of scale resolutions $\{\lambda_{\min}, \ldots, \lambda_{\max}\}$
2: **Output:** The denoising model $f_\theta$
3: **Initialize:** The model parameters $\theta$
4: **Compute** structure partition results for each $\mathcal{G} = (G^{\mathrm{sc}}, G^{\mathrm{fc}})$ using Equation 1
5: **Compute** $\mu_Q$ and $\sigma_Q$ for each scale, the mean and standard deviation of node features across defined structures in graph batch
6: **while** not converged **do**
7:     **for** $\mathcal{G}_i$ in $\{\mathcal{G}_1, \ldots, \mathcal{G}_B\}$ **do**
8:         **for** $t = 1, \ldots, T$ **do**
9:             **Sample** scalable noise $\hat{\epsilon}$ using Equation 4
10:             **Compute** loss function $\mathcal{L}$ using Equation 2
11:             **Update** model parameters $\theta \leftarrow \theta - \eta \nabla \mathcal{L}$
12:         **end for**
13:     **end for**
14: **end while**
15: **return** $f_\theta$

---

---

**Algorithm 2** Representation learning.

---

1: **Input:** Brain graph data for one subject $G^{\mathrm{sc}}, G^{\mathrm{fc}}$, forward step $\{1, \ldots, K\}$, pre-trained denoising model $f_\theta$
2: **Output:** Brain graph representation $\mathbf{Z}$
3: **for** $k$ in $\{1, \ldots, K\}$ **do**
4:     **Sample** scalable noise $\hat{\epsilon}$ using Equation 4
5:     **Compute** $G_k^{\mathrm{sc}}$ and $G_k^{\mathrm{fc}}$
6:     $\hat{\mathbf{H}}_k^{\mathrm{sc}}, \hat{\mathbf{H}}_k^{\mathrm{fc}} \leftarrow f_\theta(G_k^{\mathrm{sc}}, G_k^{\mathrm{fc}})$
7:     **Concatenate** $\mathbf{z}_k = [\hat{\mathbf{H}}_k^{\mathrm{sc}}, \hat{\mathbf{H}}_k^{\mathrm{fc}}]$
8: **end for**
9: **Concatenate** $\mathbf{Z} = [\mathbf{z}_1, \ldots, \mathbf{z}_K]$
10: **return** $\mathbf{Z}$

---

# D EXPERIMENTAL SETUP

## D.1 DETAILS OF DATASETS

**Alzheimer's disease neuroimaging initiative (ADNI)**[1]: This database originates from over 60 clinical sites across the United States and Canada, aimed at studying the manifestations of Alzheimer's disease (AD) at different stages of progression. For this study, we collected neuroimaging data, including functional magnetic resonance imaging (fMRI) and diffusion tensor imaging (DTI), from 460 participants, consisting of 211 normal controls (NC), 195 individuals with mild cognitive impairment (MCI), and 54 patients with AD. Table 3 provides detailed information about the dataset, including the participants' scores on the mini-mental state examination (MMSE) and clinical dementia rating (CDR).

**Parkinson's progression markers initiative (PPMI) Dataset**[2]: This database is collected from over 50 sites across the United States and Europe, focusing on the urgent need to identify biomarkers for Parkinson's disease (PD) onset and progression. In this study, we excluded data from repeated scans and gathered single-timepoint fMRI and DTI scans from 109 participants, including 53 healthy controls (HC) and 56 patients with PD. Table 3 summarizes this dataset, providing details about the participants and PD diagnostic criteria, including the Montreal Cognitive Assessment (MOCA) and the Unified PS Rating Scale (UPDRS).

---

[1]http://www.adni-info.org/.
[2]https://www.ppmi-info.org.

Table 3: Characteristics of Participants in ADNI and PPMI datasets

| Dataset | Type | Number | Age | MMSE | CDR |
|---------|------|--------|-----|------|-----|
| ADNI | NC | 211 | $72.8 \pm 8.3$ | $28.9 \pm 1.7$ | $0.2 \pm 0.8$ |
| | MCI | 195 | $72.8 \pm 7.9$ | $27.6 \pm 2.2$ | $1.6 \pm 1.2$ |
| | AD | 54 | $75.5 \pm 7.0$ | $22.4 \pm 2.8$ | $4.7 \pm 2.0$ |

| Dataset | Type | Number | Age | MOCA | UPDRS |
|---------|------|--------|-----|------|-------|
| PPMI | HC | 41 | $65.1 \pm 11.3$ | $27.9 \pm 2.7$ | - |
| | PD | 49 | $62.8 \pm 9.3$ | $26.9 \pm 3.7$ | $23.4 \pm 8.8$ |

## D.2 DATA PREPROCESSING

In this study, we access the fMRI and DTI data from the ADNI and the PPMI dataset. We preprocess the fMRI data using the graph theoretical network analysis (GRETNA) toolbox, based on statistical parametric mapping (SPM12) software[3]. This preprocessing included slice timing correction, head motion correction, spatial normalization, and Gaussian smoothing. The AAL atlas is used as the reference space, dividing the brain into 116 regions of interest (ROIs). Blood oxygen level-dependent (BOLD) time series corresponding to each ROI are then extracted. For DTI data, we employ the pipeline for analyzing brain diffusion images (PANDA) toolbox[4] for preprocessing, which involved skull stripping, gap cropping, motion and eddy current correction, and diffusion tensor calculation. We also use the DTI fitting tool to extract fractional anisotropy (FA) images and match them to the brain anatomical atlas template used in the fMRI data.

**Functional Brain Graph.** We construct the functional brain graph $G^{\text{fc}} = (V, \mathbf{A}^{\text{fc}}, \mathbf{X}^{\text{fc}})$ by calculating the Pearson correlation coefficients between the BOLD signals in each ROI from the preprocessed fMRI data. Here, $V = (v_1, \ldots, v_N)$ represents the node-set, where $N$ is the number of ROIs. The adjacency matrix $\mathbf{A}^{\text{fc}} \in \mathbb{R}^{N \times N}$ is derived from the Pearson correlation coefficients between pairs of nodes. Finally, $\mathbf{X}^{\text{fc}} \in \mathbb{R}^{N \times N}$ is defined as the correlation vector.

**Structural Brain Graph.** The structural brain graph $G^{\text{sc}} = (V, \mathbf{A}^{\text{sc}}, \mathbf{X}^{\text{sc}})$ is constructed from the preprocessed DTI data. Since we use the same anatomical template for both structural and functional brain networks, the definition of the node set $V$ is consistent across both graphs. To construct the graph structure, we perform local diffusion pattern reconstruction and calculate structural connectivity for each pair of nodes based on the empirical probability of fiber bundles connecting paired ROIs, resulting in the adjacency matrix $\mathbf{A}^{\text{sc}} \in \mathbb{R}^{N \times N}$. The definition of feature matrix $\mathbf{X}^{\text{sc}} \in \mathbb{R}^{N \times N}$ follows the same strategy in the functional brain graph.

## E RESULTS OF COMPARISON EXPERIMENTS

Due to text layout and page constraints, the experimental results presented in the main body focus solely on the accuracy (ACC) metric. To ensure comprehensive reporting of results including sensitivity (SEN), specificity (SPE), F1-score, and the area under the ROC curve (AUC), Table 4 and Table 5 provide the complete findings of the comparative experiments conducted on the ADNI and PPMI datasets, respectively. Notably, the most significant results are highlighted in bold, while results below the optimal threshold are underlined for clarity and emphasis.

**Analysis.** From the results across the four tables, our BrainSTORE method shows exceptional performance in both datasets. In the ADNI dataset, it achieves an average increase of 5.0% in ACC, 2.5% in SEN, 2.6% in SPE, 7.2% in F1-score, and 10.4% in AUC on the ADNI dataset. Similarly, on the PPMI dataset, we note a 3.3% increase in ACC, 5.5% in SEN, 3.4% in F1-score, and 7.6% in AUC. These results underscore the robust performance of BrainSTORE in classification tasks related to brain graph representation learning.

---

[3]https://www.fil.ion.ucl.ac.uk/spm/software/spm12/.
[4]https://www.nitrc.org/projects/panda/.

Table 4: Comparison results (%) on the PPMI dataset.

| Methods | Modality | HC vs PD | | | | |
|---|---|---|---|---|---|---|
| | | ACC | SEN | SPE | F1-score | AUC |
| BrainGNN | SC | $61.5 \pm 12.5$ | $69.1 \pm 14.9$ | $52.1 \pm 13.5$ | $66.7 \pm 8.0$ | $57.8 \pm 14.8$ |
| BrainGNN | FC | $64.6 \pm 18.4$ | $75.2 \pm 8.4$ | $67.3 \pm 8.9$ | $75.9 \pm 12.2$ | $62.6 \pm 12.9$ |
| BrainGB | SC | $66.8 \pm 7.3$ | $77.8 \pm 11.8$ | $67.9 \pm 12.3$ | $74.4 \pm 6.5$ | $59.9 \pm 10.5$ |
| BrainGB | FC | $68.4 \pm 10.7$ | $75.6 \pm 10.8$ | $56.3 \pm 11.9$ | $74.9 \pm 8.9$ | $66.9 \pm 16.6$ |
| TAN | SC,FC | $75.1 \pm 8.5$ | $79.5 \pm 13.2$ | $68.6 \pm 12.2$ | $76.6 \pm 4.5$ | $66.4 \pm 7.2$ |
| Cross-GNN | SC,FC | $84.6 \pm 7.1$ | $78.2 \pm 12.7$ | $73.6 \pm 8.4$ | $84.5 \pm 7.4$ | $82.2 \pm 8.6$ |
| RH-BrainFS | SC,FC | $\underline{85.6 \pm 7.1}$ | $\underline{84.2 \pm 12.7}$ | $\mathbf{78.6 \pm 8.4}$ | $\underline{86.4 \pm 9.6}$ | $\underline{87.2 \pm 8.6}$ |
| **BrainSTORE (ours)** | SC,FC | $\mathbf{88.9 \pm 3.4}$ | $\mathbf{89.7 \pm 4.6}$ | $\underline{75.1 \pm 6.1}$ | $\mathbf{89.8 \pm 4.2}$ | $\mathbf{94.8 \pm 7.4}$ |

Table 5: Comparison results (%) on the ADNI dataset.

(a)

| Methods | Modality | NC vs MCI | | | | |
|---|---|---|---|---|---|---|
| | | ACC | SEN | SPE | F1-score | AUC |
| BrainGNN | SC | $54.3 \pm 9.4$ | $62.9 \pm 10.4$ | $55.4 \pm 11.2$ | $57.6 \pm 11.9$ | $56.7 \pm 8.2$ |
| BrainGNN | FC | $51.8 \pm 4.3$ | $72.8 \pm 1.9$ | $61.3 \pm 6.5$ | $60.1 \pm 8.2$ | $51.9 \pm 4.3$ |
| BrainGB | SC | $55.8 \pm 4.8$ | $75.2 \pm 12.4$ | $63.8 \pm 11.2$ | $66.5 \pm 5.6$ | $58.6 \pm 4.8$ |
| BrainGB | FC | $56.6 \pm 2.5$ | $71.9 \pm 13.3$ | $60.4 \pm 7.1$ | $68.5 \pm 3.7$ | $59.2 \pm 5.9$ |
| TAN | SC,FC | $71.5 \pm 10.3$ | $67.5 \pm 5.7$ | $76.9 \pm 8.4$ | $60.6 \pm 10.5$ | $61.5 \pm 10.2$ |
| Cross-GNN | SC,FC | $\underline{82.8 \pm 6.3}$ | $86.9 \pm 7.4$ | $\underline{80.7 \pm 9.2}$ | $76.5 \pm 6.4$ | $\underline{78.4 \pm 7.2}$ |
| RH-BrainFS | SC,FC | $80.4 \pm 7.4$ | $\underline{87.4 \pm 7.1}$ | $77.6 \pm 4.2$ | $\underline{78.5 \pm 8.1}$ | $72.4 \pm 9.3$ |
| **BrainSTORE (ours)** | SC,FC | $\mathbf{85.3 \pm 6.4}$ | $\mathbf{90.2 \pm 8.3}$ | $\mathbf{82.9 \pm 5.7}$ | $\mathbf{87.9 \pm 5.4}$ | $\mathbf{83.9 \pm 8.5}$ |

(b)

| Methods | Modality | MCI vs AD | | | | |
|---|---|---|---|---|---|---|
| | | ACC | SEN | SPE | F1-score | AUC |
| BrainGNN | SC | $57.2 \pm 14.7$ | $69.7 \pm 15.9$ | $62.5 \pm 8.4$ | $66.7 \pm 18.1$ | $51.9 \pm 7.8$ |
| BrainGNN | FC | $71.1 \pm 3.7$ | $89.8 \pm 8.3$ | $72.6 \pm 3.2$ | $81.3 \pm 1.5$ | $62.0 \pm 1.1$ |
| BrainGB | SC | $74.5 \pm 6.9$ | $82.3 \pm 9.9$ | $70.4 \pm 7.2$ | $84.7 \pm 4.9$ | $63.6 \pm 4.8$ |
| BrainGB | FC | $77.3 \pm 4.7$ | $83.7 \pm 6.9$ | $76.4 \pm 5.5$ | $86.5 \pm 3.3$ | $64.2 \pm 9.6$ |
| TAN | SC,FC | $81.2 \pm 7.2$ | $90.9 \pm 10.7$ | $84.6 \pm 9.4$ | $85.6 \pm 11.4$ | $73.9 \pm 8.6$ |
| Cross-GNN | SC,FC | $83.4 \pm 6.1$ | $90.7 \pm 7.4$ | $83.6 \pm 9.3$ | $\underline{88.5 \pm 7.9}$ | $77.9 \pm 5.7$ |
| RH-BrainFS | SC,FC | $\underline{85.3 \pm 5.9}$ | $\underline{94.1 \pm 3.8}$ | $\mathbf{88.3 \pm 6.5}$ | $86.3 \pm 5.4$ | $\underline{79.4 \pm 6.9}$ |
| **BrainSTORE (ours)** | SC,FC | $\mathbf{89.4 \pm 4.7}$ | $\mathbf{95.7 \pm 5.6}$ | $\underline{86.1 \pm 6.1}$ | $\mathbf{88.6 \pm 5.8}$ | $\mathbf{91.2 \pm 5.3}$ |

(c)

| Methods | Modality | NC vs AD | | | | |
|---|---|---|---|---|---|---|
| | | ACC | SEN | SPE | F1-score | AUC |
| BrainGNN | SC | $61.7 \pm 8.5$ | $72.8 \pm 12.4$ | $60.4 \pm 7.2$ | $67.5 \pm 5.1$ | $64.8 \pm 13.8$ |
| BrainGNN | FC | $60.0 \pm 7.7$ | $56.0 \pm 12.3$ | $51.3 \pm 7.8$ | $60.3 \pm 8.7$ | $64.7 \pm 11.7$ |
| BrainGB | SC | $73.9 \pm 7.3$ | $80.5 \pm 4.8$ | $72.4 \pm 6.6$ | $73.1 \pm 4.3$ | $72.9 \pm 12.8$ |
| BrainGB | FC | $61.4 \pm 6.5$ | $65.6 \pm 5.6$ | $61.3 \pm 6.5$ | $61.3 \pm 6.5$ | $60.4 \pm 7.2$ |
| TAN | SC,FC | $75.2 \pm 9.8$ | $81.8 \pm 6.6$ | $76.0 \pm 10.7$ | $78.6 \pm 10.4$ | $\underline{75.5 \pm 11.0}$ |
| Cross-GNN | SC,FC | $80.3 \pm 8.1$ | $84.3 \pm 9.8$ | $78.9 \pm 8.4$ | $78.1 \pm 11.9$ | $70.4 \pm 10.5$ |
| RH-BrainFS | SC,FC | $\underline{82.4 \pm 7.9}$ | $\underline{89.2 \pm 9.3}$ | $\underline{83.6 \pm 6.5}$ | $\underline{80.2 \pm 7.9}$ | $74.3 \pm 7.4$ |
| **BrainSTORE (ours)** | SC,FC | $\mathbf{90.9 \pm 4.9}$ | $\mathbf{92.5 \pm 8.5}$ | $\mathbf{88.3 \pm 7.2}$ | $\mathbf{92.4 \pm 6.9}$ | $\mathbf{89.5 \pm 6.4}$ |