# OpenReview forum: "Scalable Diffusion for Bio-topological Representation Learning on Brain Graphs"
_ICLR.cc/2025/Conference — ICLR 2025 Conference Withdrawn Submission_

### Official Review · Reviewer_Yg5v · 2024-11-03

**Soundness:** 2
**Presentation:** 3
**Contribution:** 2
**Rating:** 5
**Confidence:** 4

**Summary:**

This paper proposes BrainSTORE, a framework that incorporates learning multi-modal (structural and functional) brain networks across multiple community scales to improve the classification performance of neuro/psychiatric disorders (Alzheimer’s disease and Parkinson’s disease) from MRI scans. Strengths of this work include that 1) the proposed method leverages both structural and functional connectivity, 2) it shows clear improvement over the presented baselines, and 3) the writing is generally clear and easy to follow. Some concerns and suggestions are outlined in the `Questions` section.

**Strengths:**

- The proposed method leverages both structural and functional connectivity
- The proposed method shows clear improvement over the presented baselines
- The writing is generally clear and easy to follow

**Weaknesses:**

- Extensiveness of the experiments (datasets, baseline methods)

**Questions:**

### Major concerns
- The small size of the benchmark dataset limits the reliability of the results. It would be beneficial to add experiments on datasets with larger sample sizes (e.g., HCP, at minimum).
- The number of baseline models compared seems insufficient, making it challenging to fully trust the results.
- It seems that the node feature $\mathbf{X}^{\text{fc}}$ and the adjacency matrix $\mathbf{A}^{\text{fc}}$ are identical. Please elaborate on the choice of functional connectivity as the input node feature. It should also be tested whether the GNN actually leverages the functional connectivity structure from $\mathbf{A}^{\text{fc}}$ or if it extracts connectivity information from  $\mathbf{X}^{\text{fc}}$.
- Including multi-class accuracy (ADNI) and AUC in the Table would enhance the clarity of the results.
- Further clarification of the architectural choice would improve the paper's clarity. For example, please specify the type of GNN layers, downsampling/upsampling methods for the shared backbone, etc.
- Given the direct skip connection between the early modality specific encoders and the later decoders, the effect of shared backbone network and the translation module on the performance of BrainSTORE might be trivial. Please consider adding an ablation experiment of the backbone network.


### Minor Concerns
- While the term “scale” in this paper refers to scale of the network communities here, using terms such as “scalability” or “scalable” may be misleading to the readers from the AI field, where these often imply the scale of dataset, model, or compute. Defining and elaborating the meaning of “scale” in the introduction section and substituting the words “scalable", "scalability” with terms that reflect “multi-scale” may further improve readability.
- Some minor typos can be corrected:
	- Line 328: ANDI -> ADNI
	- Line 346:  e^{-4} = 10^{-4} (confirm if learning rate is meant to be a power of ten)
	- Line 134: “that capble of” -> “that is capable of”
- In the Figures (e.g., 1 and 6), the red lines that outline the detected communities in the connectivity matrix follow only along the diagonal, which may give the impression that only nodes with adjacent indices are assigned to the same community. Adjusting this in the figures to avoid potential misinterpretation would be helpful.
- Some words or sentences in the introduction may be toned-down. For example, replacing terms like 'realistic', 'pioneers', 'authentic', 'extensive' with expressions that is more supported by the experimental results. Additionally, certain statements regarding regarding medical applications could be perceived as overstated. For example, line 038 references “supporting clinical diagnosis” and “cognitive impairment analysis,” which are not current clinical applications of the FC/SC topology.

### Recommendation
- Given the above issues, especially the extensiveness of the benchmark datasets and baseline methods, I find the novelty and impact of this work slightly underweighs the concerns. Thus, I initially recommend weak reject of the paper.

---

### Official Review · Reviewer_Rt38 · 2024-11-03

**Soundness:** 3
**Presentation:** 1
**Contribution:** 3
**Rating:** 5
**Confidence:** 5

**Summary:**

This work proposes BrainSTORE, a scalable diffusion model for brain disordering recognition. The authors present a novel idea of scalable denoising diffusion and have shown SOTA performance in disease classification and community detection. Although motivations for using diffusion and multiple modalities (SC and FC) of data are confused, and the presentation of methods is not clear, they are fixable. I will raise the score if the authors can address the weaknesses.

**Strengths:**

1. The idea of scalable denoising diffusion is novel.

2. Experimental results show BrainSTORE has SOTA performance in multiple perspectives, including classification and community detection. The visualization of multiscale community detection during denoising diffusion shows some insightful findings and aligns with current neuroscience research.

**Weaknesses:**

1. Authors overlooked the difference between BrainSTORE and expressive graph neural networks either for general purposes, e.g. GSN [1] and PathNN [2], or dedicated to brain networks, e.g., NeuroPath [3]. These methods are expressive on high-order structures with various hops, making them multi-scale. What makes BrainSTORE more advanced than those three methods in your research problems?

[1] Bouritsas, Giorgos, et al. "Improving graph neural network expressivity via subgraph isomorphism counting." IEEE Transactions on Pattern Analysis and Machine Intelligence 45.1 (2022): 657-668.

[2] Michel, Gaspard, et al. "Path neural networks: Expressive and accurate graph neural networks." International Conference on Machine Learning. PMLR, 2023.

[3] Wei, Ziquan, et al. "NeuroPath: A Neural Pathway Transformer for Joining the Dots of Human Connectomes." arXiv preprint arXiv:2409.17510 (2024).

2. The motivation of this paper can be improved.

2.1. Limitations were claimed in Sec.1, 2nd paragraph as existing methods are constrained at a single scale, while there are expressive GNNs existing such as I mentioned in Weakness#1 that are not on a single scale. Additionally, BrainGNN reviewed in this work has multiple pooling layers to merge brain regions, which is also a multi-scale method.

2.2. The authors also overlooked their motivations for using diffusion methods in BrainSTORE. Please explain why representation learning methods other than denoising diffusion methods are not better.

3. Bio-topological properties are present multiple times without a specific description. For example, in Sec.2.1, *'these methods often overlook bio-topological properties within multimodal graphs, limiting their expressiveness'* is hard to believe without a clear explanation. Do they refer to multi-scale representation? And to prevent misleading readers, does multi-scale in this work refer to various sub-graphs with different hops?

4. Figure 1 is lack of descriptions. What does the color of arrows refer to? Do the colored encoders in the bottom part of Figure 1 refer to $f_{theta}$? What is the definition of those colored decoders? What is the definition of $f_{theta}$ and Tran()? You should call back your figure when you introduce a step of the proposed methods.

5. The sample size of datasets in experiments is limited to hundreds.

6. Resolution parameter $\lambda$ controls how many scales are learned to brain network representation. I suggest authors show the trend of accuracy when altering the range of $\lambda$ to evaluate their ideas.

7. I suggest authors show the ablation studies of multimodal input by solely using SC or FC.

**Questions:**

1. Which dataset is used in Table 2?

2. What are the data pre-processing steps? How do authors deal with the negative FC?

---

### Official Review · Reviewer_PTs1 · 2024-11-04

**Soundness:** 3
**Presentation:** 3
**Contribution:** 3
**Rating:** 5
**Confidence:** 4

**Summary:**

This paper proposes a novel Scalable diffusion model for bio-TOpological REpresentation learning on Brain graphs called BrainSTORE, which constructs multiscale topological structures within brain graphs, facilitating a deep exploration of biotopological properties.

**Strengths:**

1. The paper models biologically multiscale topological structures to learn the patterns in brain graphs.
2. The paper conducts extensive experiments on two multimodal brain disease datasets to validate the effectiveness of the proposed BrainSTORE.

**Weaknesses:**

1. Some experiments are missing. How about the classification result of BrainSTORE by using a single modality like SC or FC? The baselines including BrainGNN and BrainGB only use a single modality, so it is hard to show that the better performance of BrainSTORE is due to the method itself or more modalities information.
2. In the visualization of classification results in Fig3, the ADNI has three classes, but the figure only shows two clusters, what does the color represent? HC, MCI, or AD? How about the result in PPMI dataset.
3. Fig4. shows the performance of the model using the scalable learning strategy. How do you replace it with a different algorithm?
4. The result of ADNI in Fig 4 and Fig 5 shows the single accuracy, but it conducts three binary classifications in table 1, do you conduct one vs one classification here? Table 2 should also shows the result of two datasets.
5. In explanation of community detection, the model assigned the nodes to different communities, is there any reference to support that some specific nodes belong to the same or different communities in patients from the clinical perspective?

**Questions:**

see above. The questions focus on some descriptions of the results and the additional experimental results in two datasets.

---

### Official Review · Reviewer_pgFm · 2024-11-04

**Soundness:** 2
**Presentation:** 2
**Contribution:** 2
**Rating:** 3
**Confidence:** 4

**Summary:**

The authors of this paper proposed BrainSTORE, which is a novel diffusion model for multimodal graph by integrating topological features across modalities. It captures multi-modal and multi-scale brain networks within a unified framework to utilized enhanced feature representations for graph classification task. As a result, the proposed framework outperformed recent graph neural networks on two neuroimaging datasets.

**Strengths:**

- The motivation and idea to integrate multi-scale and multi-modal features across brain networks is interesting and novel.
- Based on the Tables 1 and 4, the proposed method mostly outperformed multiple recent baseline methods. All experiments were performed with 10-fold cross validation to avoid biased results.

**Weaknesses:**

- In line 72, the authors claimed that the proposed method addresses ‘authentic’ bio-topological representations. I would like to ask why authors used the term ‘bio-topology’ instead of ‘topology’ and ask how they could capture the authentic bio-topological features. I assume that the bio-topological feature should reflect information derived from biologically meaningful topologies (e.g., brain networks defined on pre-defined atlases), while the topological features might represent features from any arbitrary types of brain parcellations. If this interpretation is correct, multi-scale bio-topology would then correspond to multiple brain networks defined on multiple pre-defined atlases (or other schemes that involve biological meanings). However, in this paper, they are simply set as graphs calculated using a hyperparameter $\lambda$, which may not sufficiently capture biologically meaningful structures.

- In line 755, the authors explained that they identified an ‘optimal’ set of scales $lambda = [0.5, 1.5]$, but it is unclear how they obtained these values. The authors explained that they used the knowledge of brain functional modules to set the scales, but providing much detailed explanation would be helpful. Which knowledge exactly do you used and how you applied it to set the parameter?

- Also, as the number of denoising step was set to 3, I understand that only three scales (i.e., 0.5, 1.0, and 1.5) were used to capture multi-scale features during denoising. I concern that this range of scales may be under-examined, and only using three scales may miss features from other resolution which could reflect critical topological information.

- The reason of setting the key parameters $\tau$ and $K$ to 0.5 and 1.0, respectively, are quite unclear to me. I assume that the maximum values of these parameters are not necessarily to be set as 1, and could be larger than 1. Also, setting the parameters identical across all nodes highly likely limit the flexibility to capture localized variations across nodes. In the appendix B, authors explained that a large $\tau$ may result in complicate community detection, but based on the Fig. 7, a smaller $\tau$ (Fig. 7a) seems resulting in more complicated communities across scales. Also, based on the same figure, the $\tau$ and scales $\lambda$ are dependent each other, which may need to set different $tau$ for different scale.

- Minor weakness: There are many typos, e.g., Vallina (Vanilla is correct) diffusion model in line 115. In line 132, the $i$ and $j$ should range from 1 to $N$ as the number of nodes is $N$. In line 748, ‘Figure 9’ needs to be replaced to ‘Figure 8’. In Table 1, ‘ANDI’ needs to be replaced to ‘ADNI’.

**Questions:**

- It would be helpful if the authors could provide clarification on why the node feature is set as a set of connectivity correlations, and provide any references that used this setting for graph analyses.
- Why didn’t you perform 3-way classification for the ADNI dataset? Is there any reason to perform three binary classifications instead of a single three-way classification (i.e., NC vs. MCI vs. AD)?

---

### Note · Authors · 2024-11-16

I have read and agree with the venue's withdrawal policy on behalf of myself and my co-authors.